# Determination of Single-Ion Partition Coefficients between Water and Plasticized PVC Membrane Using Equilibrium-Based Techniques

**DOI:** 10.3390/membranes12101019

**Published:** 2022-10-20

**Authors:** Andrei V. Siamionau, Vladimir V. Egorov

**Affiliations:** 1Laboratory of the Physical Chemical Investigation Methods, Research Institute for Physical Chemical Problems of the Belarusian State University, Leningradskaya Str., 14, 220006 Minsk, Belarus; 2Analytical Chemistry Department, Faculty of Chemistry, Belarusian State University, Leningradskaya str., 14, 220030 Minsk, Belarus

**Keywords:** single-ion partition coefficients, coextraction constants, association constants, exchange constants, selectivity coefficients, potentiometry, ion-selective electrodes

## Abstract

An experimentally simple method for the direct determination of single-ion partition coefficients between water and a PVC membrane plasticized with o-NPOE is suggested. The method uses the traditional assumption of equal single-ion partition coefficients for some reference cation and anion, in this case tetraphenylphosphonium (TPP^+^) and tetraphenylborate (TPB^−^). The method is based on an integrated approach, including direct study of some salts’ distribution between water and membrane phases, estimation of ion association constants, and measurements of unbiased selectivity coefficients for ions of interest, including the reference ones. The knowledge of distribution coefficients together with ion association constants allows for direct calculation of the multiple of the single-ion partition coefficients for the corresponding cation and anion, while the knowledge of unbiased selectivity coefficients together with ion association constants allows for immediate estimation of the single-ion partition coefficients for any ion under study, if the corresponding value for the reference ion is known. Both potentiometric and extraction studies are inherently equilibrium-based techniques, while traditionally accepted methods such as voltammetry and diffusion are kinetical. The inner coherent scale of single-ion partition coefficients between water and membrane phases was constructed.

## 1. Introduction

The concept of a single-ion partition coefficient first introduced by Eisenman [1] is widely used in the theoretical description of the functioning of ion-selective electrodes (ISEs) and optodes [2,3,4,5,6,7,8,9,10,11,12,13,14,15,16,17,18]. According to [1], a single-ion partition coefficient depends on standard Gibbs energies of hydration and solvation. It corresponds to a hypothetical ratio of ion activities in corresponding phases, provided that the boundary electric potential is zero. However, single-ion partition coefficients are experimentally difficult to obtain and, moreover, to the best of our knowledge, up to now there has not been any attempt to experimentally determine single-ion partition coefficients between water and plasticized polyvinyl chloride (PVC) membranes, which are commonly used in practice. The data of [19], where single-ion partition coefficients between water and PVC–o-NPOE mixture were presented, cannot be attributed to typical ion-selective electrode (ISE) membranes, since the content of PVC in the composition studied was only 2.8%. Bakker et al. suggested a novel approach to estimate the Na^+^ and Cl^−^ single-ion partition coefficient product between water and plasticized PVC membrane via voltammetry [20]; however, the obtained value of 10^−9.6^ was found to be 4 orders higher than that estimated from the upper detection limit (UDL) of a potassium-selective electrode membrane that contained valinomycin as an ionophore and the same macrocomponents, i.e., PVC and o-NPOE [21].

At the same time, there is a large quantity of literature data comprising single-ion partition coefficients between water and organic solvents, e.g., nitrobenzene, 1,2-dichloroethane (DCE), and o-NPOE, obtained by different methods, such as partition [22,23], solubility measurements [24,25], voltammetric study of transfer across the water/solvent interface [23,24,25,26,27,28,29,30,31], and transport studies across supported liquid membranes [32]. Computational approach [33] was also applied. However, the literature data are often inconsistent with each other. Even for reference ions (tetraphenylarsonium (TPAs^+^) and tetraphenylborate (TPB^−^)) the single-ion partition coefficient values cited in different sources in some cases differ by more than ten times (for instance [25,32]), while the difference in these values for more hydrophilic ions such as sodium and nitrate is 2.5 orders of magnitude [28,32]. It should be noted that the similar situation takes place for complexation constants in the membrane phase. For example, the complex formation constant for potassium with valinomycin determined by voltammetry [19] and potentiometry [34] differs by 4.4 orders of magnitude.

To date, the vast majority of the single-ion partition coefficient values between water and organic solvents have been obtained using an indirect method, namely, voltammetric study of transfer across the water/solvent interface. However, as noted in [35], this is an intrinsically kinetic method, so a number of systematic errors may be present, compared with potentiometric or solvent extraction techniques. In particular, the potential drop in the organic phase, the diffusion coefficients’ uncertainty, and the ion association in the organic phase can result in the significant deterioration of experimental results. Hence, experimental methods that depart less from equilibrium would be more appropriate for these aims. It should be noted, however, that solubility measurements are inappropriate for studying, either very poorly or very well, soluble salts or those forming crystal hydrates, and they do not take into account the ion association.

Therefore, the problem of the reliable estimation of single-ion partition coefficients between water and plasticized PVC membranes for widespread practically important ions is rather demanding and topical. The most rational way to solve this problem seems to be a combination of partition measurements and potentiometric techniques, since both methods directly interrogate the equilibrium state. The present article critically analyzes the known literature data on the single-ion partition coefficients between water and organic solvents. In addition, it first suggests a complex approach for their estimation that comprises experimental determination of salts’ distribution coefficients between aqueous solutions and one of the most commonly used membrane compositions containing 66.7% o-NPOE and 33.3% PVC, the measurements of unbiased selectivity coefficients, and the determination of ion association constants from both extraction and potentiometric data.

## 2. Brief Theoretical Background

In order to estimate the single-ion partition coefficients between phases, it is necessary to know the coextraction constant value of at least one salt Cat^+^An^−^ and exchange constants for the series of cations and anions including reference ions TPP^+^ and TPB^−^. The coextraction constant is defined as a product of the corresponding single-ion partition coefficients: $K_{\mathrm{ex}}({\mathrm{Cat}}^{+}{,\mathrm{An}}^{-})\equiv k_{{\mathrm{Cat}}^{+}}\cdot k_{{\mathrm{An}}^{-}}$, where $k_{{\mathrm{Cat}}^{+}},\;\;\;k_{{\mathrm{An}}^{-}}$ are single-ion partition coefficients of the cation and anion that depend upon free Gibbs energies of transfer from aqueous to membrane phase $k_{i}\equiv \exp\left\{\left[{\left(\Delta G_{i}^{o}\right)}_{\mathrm{aq}}-{\left(\Delta G_{i}^{o}\right)}_{m}\right]/RT\right\}$, where i is any cation or anion under study. The ion exchange constant is defined as a ratio of the corresponding single-ion partition coefficients. For instance, $K_{{\mathrm{Cat}}^{+}{,\mathrm{TPP}}^{+}}^{\mathrm{Exch}}\equiv k_{{\mathrm{TPP}}^{+}}/k_{{\mathrm{Cat}}^{+}};\;\;\;K_{{\mathrm{An}}^{-}{,\mathrm{TPB}}^{-}}^{\mathrm{Exch}}\equiv k_{{\mathrm{TPB}}^{-}}/k_{{\mathrm{An}}^{-}}$. If the corresponding coextraction and ion exchange constants are known, the coextraction constant of the salt formed by the reference ions, namely, TPP^+^TPB^−^, can be calculated by Equation (1) as its high value makes direct experimental measurement complicated:(1)$$\log K_{\mathrm{ex}}{(\mathrm{TPP}}^{+}{,\mathrm{TPB}}^{-})=\log K_{\mathrm{ex}}{(\mathrm{Cat}}^{+}{,\mathrm{An}}^{-})+\log K_{{\mathrm{Cat}}^{+}{,\mathrm{TPP}}^{+}}^{\mathrm{Exch}}+\log K_{{\mathrm{An}}^{-}{,\mathrm{TPB}}^{-}}^{\mathrm{Exch}}$$

Further, the single-ion partition coefficients of the reference ions TPP^+^ and TPB^−^ can be calculated by Equation (2), and for any other cation or anion, by Equations (3) and (4), respectively:(2)$$\log k_{{\mathrm{TPP}}^{+}}\equiv \log k_{{\mathrm{TPB}}^{-}}\equiv \frac{\log K_{\mathrm{ex}}{(\mathrm{TPP}}^{+}{,\mathrm{TPB}}^{-})}{2}$$
(3)$$\log k_{{\mathrm{Cat}}^{+}}=\log k_{{\mathrm{TPP}}^{+}}-\log K_{{\mathrm{Cat}}^{+}{,\mathrm{TPP}}^{+}}^{\mathrm{Exch}}$$
(4)$$\log k_{{\mathrm{An}}^{-}}=\log k_{{\mathrm{TPB}}^{-}}-\log K_{{\mathrm{An}}^{-}{,\mathrm{TPB}}^{-}}^{\mathrm{Exch}}$$

It was convincingly proven earlier using the sandwich membrane technique that, in PVC membranes plasticized with o-NPOE ion-pairing for quaternary ammonium cations, containing four long-chain substituents at the nitrogen atom with different anions (from chloride to picrate) can be neglected [36]. Hence, the quaternary ammonium salts are quite convenient objects for the determination of experimental coextraction constants as they can be directly calculated from the distribution experiments. In the present work, tetrabutylammonium bromide, which has hydrophobicity acceptable for experimental studies and does not form ion associates (at least up to 10^−2^ M concentration in PVC membrane plasticized with o-NPOE), was used for this purpose. Furthermore, the formation of ion associates can be completely neglected in the case of anion-exchange membranes based on long-chain quaternary ammonium salts, which simplifies the determination of the anion-exchange constants to the well-known routine procedure of measuring unbiased selectivity coefficients [37]:(5)$$K_{An_{1}^{-},An_{2}^{-}}^{Exch}\equiv \frac{k_{An_{2}^{-}}}{k_{An_{1}^{-}}}=K_{An_{1}^{-},An_{2}^{-}}^{Pot}$$
where $K_{An_{1}^{-},An_{2}^{-}}^{Pot}$ is an unbiased selectivity coefficient.

In the case of cations, the determination of exchange constants is substantially complicated as the formation of ion associates cannot be neglected. According to [38], salts of one of the most popular cation exchangers, namely, *tetrakis-*(4-chlorophenyl)borate with alkali metals are associated to about 80% for traditionally used concentrations in the membrane of around 1 × 10^−2^ M.

If both ions and ion pairs are present in the membrane, then the material balance equation in the membrane phase for either an electrolyte CatAn as a whole or one of its parts (e.g., Cat^+^ or An^−^) reads:(6)$$\begin{array}{l} \bar{C_{Kat,An}^{tot}}=\bar{C_{Kat^{+}}^{tot}}=\bar{C_{An^{-}}^{tot}}=\\ =\sqrt{K_{ex}(Cat^{+},An^{-})\cdot a_{Cat^{+}}\cdot a_{An^{-}}}+{\left(k_{ass}\right)}_{Cat^{+}\ldots An^{-}}\cdot K_{ex}(Cat^{+},An^{-})\cdot a_{Cat^{+}}\cdot a_{An^{-}} \end{array}$$
where $\bar{C_{Cat,An}^{tot}}$, $\bar{C_{Cat^{+}}^{tot}}$, $\bar{C_{An^{-}}^{tot}}$ are total concentrations of electrolyte CatAn, and its components Cat^+^ and An^−^ in the membrane phase, respectively; $a_{Cat^{+}}$ and $a_{{\mathrm{An}}^{-}}$ are activities of the corresponding ions in the solution, ${\left(k_{\mathrm{ass}}\right)}_{{\mathrm{Cat}}^{+}\ldots {\mathrm{An}}^{-}}$ is the ion association constant of Cat^+^…An^−^, $K_{ex}(Cat^{+},An^{-})$ is the coextraction constant. The first summand in Equation (6) describes the free ions’ concentration in the membrane, while the second characterizes the ion associates’ concentration.

If the equilibrium concentrations of the extracted salt in the membrane phase differ broadly enough in a series of experiments, the coextraction and association constants can be estimated simultaneously from the distribution data. The calculated values of cation or anion total concentration in the membrane and their activities in aqueous solution can be substituted in Equation (6). Then, the system of such equations can be resolved for two different concentrations in the membrane phase. More reliable results can be obtained if a fitting method is used as it takes into account the variety of experimental data (see also Appendix A). Since the direct study of salts’ distribution between solution and plasticized membrane is rather laborious, time consuming, and applicable for a limited number of substances with moderate values of coextraction constants, the potentiometric methods for determination of ion association constants were also implemented (see Appendix B).

The knowledge of ion association constants allows for calculation of the free ion concentration in the membrane phase and determination of the exchange constant values from unbiased selectivity coefficients, using the well-known equation that describes $K_{{\mathrm{Cat}}_{1}^{+}{,\mathrm{Cat}}_{2}^{+}}^{\mathrm{Pot}}$ as a function of single-ion partition coefficients and free ion concentrations in the membrane phase [39]. In the considered case it takes the following form:(7)$$K_{{\mathrm{Cat}}_{1}^{+}{,\mathrm{Cat}}_{2}^{+}}^{\mathrm{Exch}}\equiv \frac{k_{{\mathrm{Cat}}_{2}^{+}}}{k_{{\mathrm{Cat}}_{1}^{+}}}=K_{{\mathrm{Cat}}_{1}^{+}{,\mathrm{Cat}}_{2}^{+}}^{\mathrm{Pot}}\cdot \frac{{\bar{C}}_{{\mathrm{Cat}}_{2}^{+}}}{{\bar{C}}_{{\mathrm{Cat}}_{1}^{+}}}$$

## 3. Materials and Methods

**Reagents.** Cocktails for ISE membranes were prepared using high molecular mass PVC as the polymeric matrix and o-NPOE as the plasticizer (both from Selectophore^®^, Fluka, Steinheim, Germany). Sodium *tetrakis*-(4-chlorophenyl)borate (NaTClPB), potassium *tetrakis*-(4-chlorophenyl)borate (KTClPB), and tetradodecylammonium chloride (TDDACl) used as electroactive compounds were from Selectophore^®^, Fluka, and sodium tetraphenylborate (NaTPB) was from Alfa Aesar (Karlsruhe, Germany). Trinonyloctadecylammonium bromide (TNODABr) and tetradecylammonium nitrate (TDANO_3_) were synthesized and purified at the Analytical Chemistry Department of Belarusian State University (Minsk, Belarus) according to [40,41]. The content of amine impurities did not exceed 0.1%. All salts used for the preparation of solutions were of puriss grade. Tetrahydrofuran (THF, Belreahim, Minsk, Belarus) was distilled before use.

**The TBABr coextraction constant determination**. Membranes containing PVC and o-NPOE in a 2 to 1 ratio, without ion exchangers, were cut into small pieces, immersed in TBABr solutions (from 10^−3^ to 5 × 10^−2^ M) containing 1 M or 0.1 M NaBr, and stirred for 3 days. To establish the concentration of TBABr in the membrane phase, the quantitative re-extraction of the substance from the membrane into the aqueous phase was carried out by several repeated treatments with distilled water. The equilibrium TBABr concentration in initial and re-extraction water solutions was determined by the potentiometric titration if the concentration was higher than 5 × 10^−3^ M. For lower concentrations, direct potentiometry using a TBA^+^-selective electrode was applied (See Appendix C).

The coextraction constant of TBABr was calculated based on the assumption of complete dissociation in the membrane phase using the formulae:(8)$$K_{\mathrm{ex}}{(\mathrm{TBA}}^{+}{,\mathrm{Br}}^{-})=\frac{{\left(\bar{C_{\mathrm{TBABr}}^{\mathrm{tot}}}\right)}^{2}}{a_{{\mathrm{TBA}}^{+}}\cdot a_{{\mathrm{Br}}^{-}}}$$
where $K_{\mathrm{ex}}{(\mathrm{TBA}}^{+},\;{\mathrm{Br}}^{-})$ is the TBABr coextraction constant, $a_{{\mathrm{TBA}}^{+}}$ and $a_{{\mathrm{Br}}^{-}}$ are the corresponding ions activities in the solution phase, $\bar{C_{\mathrm{TBABr}}^{\mathrm{tot}}}$ is the equilibrium TBABr concentration in the membrane phase.

The activity coefficients of bromide ion in concentrated solutions were taken from [42], and the TBA^+^ ion activity coefficients were determined from the potential measurements in 10^−4^ M TBABr solutions, prepared using distilled water and the NaBr background of the corresponding concentration (see Appendix D). The absence of a distinct dependence of the extraction constant, calculated according to Equation (8), upon equilibrium concentration in the membrane phase proved the validity of the complete dissociation assumption in the case of TBABr.

**The TPPCl and NaTPB coextraction constants’ determination**. The extraction studies were performed in the same fashion as previously described; however, due to the better affinity of TPPCl and NaTPB to the membrane phase, no background electrolyte was used. After equilibrium had been established, TPPCl and NaTPB were quantitatively re-extracted into the water phase by several re-treatments with distilled water. The TPPCl content in the re-extraction solutions was determined similarly to the TBABr. The NaTPB content was determined by the precipitation titration using TBABr as titrant.

It was found that the TPPCl and NaTPB coextraction constants calculated by the formula analogues to Equation (8) distinctly increase as the equilibrium salt concentration in the membrane phase increases, which is evidence of the ion association in the membrane phase. In this case, total salt concentration in the membrane phase is described by Equation (6), and both coextraction and ion association constants can be estimated directly from the distribution studies by solving a system of equations for a pair of experimental points. It is important that the difference between measured concentrations in the membrane phase for the selected pair of the experimental points was not two small. The results obtained for several pairs of points were averaged.

Different fitting methods were also implemented. In particular, the next expression for the coextraction constant follows from Equation (6):(9)$$K_{\mathrm{ex}}{(\mathrm{Cat}}^{+}{,\mathrm{An}}^{-})=\frac{{\left[-1+\sqrt{1+4\bar{C_{\mathrm{Cat},\mathrm{An}}^{\mathrm{tot}}}\cdot {\left(k_{\mathrm{ass}}\right)}_{{\mathrm{Cat}}^{+}\ldots {\mathrm{An}}^{-}}}\right]}^{2}}{4{\left[{\left(k_{\mathrm{ass}}\right)}_{{\mathrm{Cat}}^{+}\ldots {\mathrm{An}}^{-}}\right]}^{2}\cdot a_{{\mathrm{Cat}}^{+}}\cdot a_{{\mathrm{An}}^{-}}}$$

The association constant value was fitted into the experimental data set to minimize mean square deviation of coextraction constants. Obtained $K_{\mathrm{ex}}{(\mathrm{TPP}}^{+},\;{\mathrm{Cl}}^{-})$ and $K_{\mathrm{ex}}{(\mathrm{Na}}^{+},\;{\mathrm{TPB}}^{-})$ values were averaged (see also Appendix A).

**Potentiometric studies.** The measurements were performed using an Ecotest-120 ionometer (Econix, Moscow, Russia) and an Ag/AgCl reference electrode EVL-1M3.1 (Izmeritel, Gomel, Belarus) filled with 3.5 M KCl solution. For performing measurements in perchlorate solutions, the reference electrode was filled with equally transferring solutions, containing 0.318 M NaCl and 0.341 M Na_2_SO_4_ in order to prevent the formation of potassium perchlorate precipitate in the capillary. All measurements were performed under constant stirring and temperature 291 ± 2 K. The diffusion potential was accounted for using Henderson formulae.

**Electrode preparation.** All components (PVC, plasticizer, ion exchanger) were weighed precisely, dissolved in THF, and stirred for 2 h. The solution was poured into the glass ring fixed on the glass plate and the solvent was allowed to evaporate overnight. Membrane disks 1 mm thick and 11 mm in diameter were cut from the parent membrane and glued on the top of the poly(methyl methacrylate) tubes with PVC-THF composition (1 g of PVC per 11 mL of THF).

**Determination of unbiased selectivity coefficients** was performed according to [37], by the potential determination in solutions of different concentrations containing corresponding ions, in order from the most hydrophilic (Cl^−^ in the case of anions and Li^+^ in the case of cations) to the most extractable ion (TPB^−^ and TPP^+^ respectively) (see Appendix E). Calculated $K_{i,j}^{Pot}$ values were equated to ion exchange constants in the case of anions, or were used to calculate the ion exchange constants of cations investigated for TPP^+^ according to Equation (7). The free TBA^+^ and tetraethylammonium (TEA^+^) ion concentrations were assumed to be equal to the total ion exchanger concentration in membrane phase $\bar{C_{R}^{tot}}$; for all the other cations $\bar{C_{Cat_{1}^{+}}}$ values were calculated from Equation (10) using specially determined ion association constants of the corresponding cations with TClPB^−^ anion (see Appendix B):(10)$$\bar{C_{{\mathrm{Cat}}^{+}}}+{\left(k_{\mathrm{ass}}\right)}_{{\mathrm{Cat}}^{+}\ldots {\mathrm{TClPB}}^{-}}\cdot {\left(\bar{C_{{\mathrm{Cat}}^{+}}}\right)}^{2}=\bar{C_{R}^{\mathrm{tot}}}$$

## 4. Results and Discussion

It follows from Equation (6) that, in the ultimate case when the free ions dominate and the presence of ion associates can be neglected, the dependence of the concentration of the extracted substance in the membrane upon the product of ions activities in water phase in coordinates log $\bar{C_{\mathrm{Kat},\mathrm{An}}^{\mathrm{tot}}}$ − log($a_{{\mathrm{Kat}}^{+}}$·$a_{{\mathrm{An}}^{-}}$) should be linear and the tangent is close to 0.5. On the contrary, if the substance is primarily present in the form of ion associates, the tangent should be close to 1 in the same coordinates. It follows from Figure 1 that the first case corresponds to the TBABr extraction, which is consistent with [36] and justifies the assumption of complete dissociation of quaternary ammonium salts. The value of log*K*_ex_(TBA^+^,Br^−^), calculated from the data depicted in the Figure 1 bi-logarithmic dependence, was −2.55 ± 0.10.

In case of TPPCl and NaTPB, the tangent is close to 1, which suggests strong ion association in the membrane phase. Obviously, the impact of the first summand in the right side of Equation (6) in comparison with the second decreases with an increase in the product of ion activities. The presence of both free ions and ion associates in the membrane phase suggests the following inequality:(11)$$K_{ex}(Cat^{+},An^{-})<\frac{{\left(\bar{C_{\mathrm{Cat},\mathrm{An}}^{\mathrm{tot}}}\right)}_{\min}^{2}}{{\left(a_{{\mathrm{Cat}}^{+}}\cdot a_{{\mathrm{An}}^{-}}\right)}_{\min}}$$
where subscript “min” refers to the minimal experimental values of the electrolyte equilibrium concentration in the membrane and the product of ion activities in the water phase.

Therefore, there is a simple opportunity of an approximate estimation of the upper limit value of the coextraction constant from the experimental data set. According to the experimental conditions, the minimal concentrations of TPPCl and NaTPB in the membrane phase were 3.16 × 10^−4^ and 1.37 × 10^−3^, while the products of ion activities were 9.18 × 10^−7^ and 1.75 × 10^−5^, respectively. Hence, in case of TPPCl and NaTPB, the values of the coextraction constants should not exceed 1.1·10^−1^ for both substances.

On the other hand, when the concentration of the components in the membrane phase is high and ion associates dominate over free ions, the lower limit value of the association constant can be estimated using the previously calculated (and in this case definitely overestimated) value of the coextraction constant by implementing the following inequality:(12)$${\left(k_{ass}\right)}_{{\mathrm{Cat}}^{+}\ldots {\mathrm{An}}^{-}}>\frac{{\left(\bar{C_{\mathrm{Cat},\mathrm{An}}^{\mathrm{tot}}}\right)}_{\max}}{K_{\mathrm{ex}}{(\mathrm{Cat}}^{+}{,\mathrm{An}}^{-})\cdot {\left(a_{{\mathrm{Cat}}^{+}}\cdot a_{{\mathrm{An}}^{-}}\right)}_{\max}}$$
where $K_{ex}(Cat^{+},\;An^{-})$ is the upper limit value of the coextraction constant estimated using Inequality (11), subscript “max” refers to the maximal experimental values of the total equilibrium concentration of the corresponding electrolyte in the membrane and the product of ion activities in the water phase, and ${\left(k_{ass}\right)}_{Cat^{+}\ldots An^{-}}$ is the lower limit of the possible association constant value. According to the experimental results, the values of ${\left(k_{ass}\right)}_{Cat^{+}\ldots An^{-}}$ are no less than 1.0·10^4^ for TPPCl and 9.3·10^2^ for NaTPB.

According to Equation (6), the experimental data were fitted to the following values of ion association constants: ${\left(k_{ass}\right)}_{TPP^{+}\ldots Cl^{-}}$ = (4.5 ± 1.8)·10^4^, ${\left(k_{ass}\right)}_{Na^{+}\ldots TPB^{-}}$ = (1.6 ± 0.4)·10^4^, which is completely consistent with the a preliminary estimation. In addition, the ${\left(k_{ass}\right)}_{Na^{+}\ldots TPB^{-}}$ value is in satisfactory agreement with the value of (1.1 ± 0.1)·10^4^, estimated from potentiometric data (see Appendix B). The corresponding coextraction constant values estimated from the distribution data are (5.1 ± 2.0)·10^−3^ for TPPCl and (4.2 ± 1.3)·10^−3^ for NaTPB.

Table 1 presents unbiased selectivity coefficients for different cations and experimentally determined anions versus reference ions (TPP^+^ and TPB^−^) and ion-exchange constants estimated from potentiometric data (see Appendix A and Appendix B). Both the selectivity coefficients and association constants are obtained with sufficient accuracy so that they can be exploited to estimate corresponding ion exchange constants according to Equation (7). Simultaneous solving of Equations (7) and (10) results in the following expression for the ion exchange constant:(13)$$K_{{\mathrm{Cat}}^{+}{,\mathrm{TPP}}^{+}}^{\mathrm{Exch}}=K_{Cat^{+},TPP^{+}}^{Pot}\cdot \frac{\sqrt{1+4{\bar{C}}_{R}^{\mathrm{tot}}\cdot {\left(k_{\mathrm{ass}}\right)}_{{\mathrm{TPP}}^{+}{\mathrm{TClPB}}^{-}}}-1}{\sqrt{1+4{\bar{C}}_{R}^{\mathrm{tot}}\cdot {\left(k_{\mathrm{ass}}\right)}_{{\mathrm{Cat}}^{+}{\mathrm{TClPB}}^{-}}}-1}\cdot \frac{{\left(k_{\mathrm{ass}}\right)}_{{\mathrm{Cat}}^{+}{\mathrm{TClPB}}^{-}}}{{\left(k_{\mathrm{ass}}\right)}_{{\mathrm{TPP}}^{+}{\mathrm{TClPB}}^{-}}}$$

In order to estimate the single-ion partition coefficients, the non-thermodynamic assumption of the free Gibbs transfer energies from water in an organic solvent equality for TPB^−^ and TPAs^+^ (Parker assumption [43]), or similarly for TPB^−^ and TPP^+^ [32,33,44], is usually taken. The obtained experimental data can be used to calculate the coextraction constant of TPP^+^ and TPB^−^ in a number of ways, starting from coextraction constants of TPPCl, TBABr, or NaTPB, and the corresponding ion exchange constants:$$\log K_{\mathrm{ex}}{(\mathrm{TPP}}^{+}{,\mathrm{TPB}}^{-})=\log K_{\mathrm{ex}}{(\mathrm{TPP}}^{+}{,\mathrm{Cl}}^{-})+\log K_{{\mathrm{Cl}}^{-}{,\mathrm{TPB}}^{-}}^{\mathrm{Exch}}=-2.28+13.28=11.00$$
$$\begin{array}{l} \log K_{\mathrm{ex}}{(\mathrm{TPP}}^{+}{,\mathrm{TPB}}^{-})=\log K_{\mathrm{ex}}{(\mathrm{TBA}}^{+}{,\mathrm{Br}}^{-})+\log K_{{\mathrm{TBA}}^{+}{,\mathrm{TPP}}^{+}}^{\mathrm{Exch}}+\log K_{{\mathrm{Br}}^{-}{,\mathrm{TPB}}^{-}}^{\mathrm{Exch}}=\\ -2.55+1.46+11.83=10.74 \end{array}$$
$$\log K_{\mathrm{ex}}{(\mathrm{TPP}}^{+}{,\mathrm{TPB}}^{-})=\log K_{\mathrm{ex}}{(\mathrm{Na}}^{+}{,\mathrm{TPB}}^{-})+\log K_{{\mathrm{Na}}^{+}{,\mathrm{TPP}}^{+}}^{\mathrm{Exch}}=-2.37+13.19=10.82$$

The results of the above calculations justify inner consistency, coherency, and reliability of the experimental data. Despite the good match of all the calculated values, the most reliable is $\log K_{\mathrm{ex}}{(\mathrm{TPP}}^{+}{,\mathrm{TPB}}^{-})=10.74$, computed from the TBABr coextraction constant, as the extraction process in this case was not aggravated with the ion association in the membrane phase and the experimental data set contained the maximal number of points. Hence, the logarithmic values of the single-ion partition coefficients for TPP^+^ and TPB^−^ are taken equal to 5.37. The $\log k_{i}$ values for all other ions calculated according to Equations (3) and (4) are presented in Table 2.

It can be seen from Table 2 that the vast majority of experimental data for the single-ion partition coefficients between water and o-NPOE are obtained for relatively hydrophobic cations and anions, while the data for cations, which are more hydrophilic than cesium, and anions, which are more hydrophilic than perchlorate, are present only in two works. In some cases, the data from different papers do not match each other. For example, the *k*_i_ value for TMA^+^, obtained in [29], differs by approximately by 5 orders of magnitude from the corresponding values in other works, which agree with each other within 0.5 logarithmic units. If the typo in the text [29] is assumed and the true value has an opposite sign, the difference with other works is still substantial, i.e., about 1.5 orders of magnitude. Next, the *k*_i_ values for the TPB^−^ reference ion are consistent (within 0.2 logarithmic units), except for that of paper [32]. In addition, the literature data analysis reveals some inner inconsistency, even for the results obtained in one particular work. For example, the *k*_i_ values for TEA^+^ and TMA^+^, determined in [29], are not consistent with the general logic of hydrophobicity dependence upon the hydrocarbon substituent length. One more example is the values of the single-ion partition coefficients given in [28] and [32], which differ by more than an order of magnitude for iodide and more than 2.5 orders of magnitude for nitrate. According to [32], I^−^ and $NO_{3}^{-}$ ions are approximately equally extractable, which contradicts experimental results on ion-exchange constants and unbiased selectivity coefficients for electrodes based on QAS [45]. According to [45], the iodide ion is 1.3 orders of magnitude more lipohilic than the nitrate ion. The last value corresponds well to the differences in single-ion partition coefficients for iodide and nitrate equal to 1.6 and 1.5 logarithmic units, obtained in [28] and in the present paper, respectively.

Among all the available data, the results presented in [28] seem the most coherent and reliable. The values determined in the present work are in general consistent with the data of [28]. It is worth noting that the single-ion partition coefficient values for the pure o-NPOE and plasticized PVC membrane should, in principle, differ from each other as the solvating properties of the media are different. Although the electron-donating ability of o-NPOE is rather weak, it is nevertheless higher than that of chlorinated hydrocarbons, which in this respect can be considered analogues of PVC (according to [46] the donor number is 10 for nitrobenzene and 8 for 1,2-dichloroethane). Therefore, solvating properties of a plasticized PVC membrane relative to hydrophilic cations are weakened in comparison with the solvating properties of pure o-NPOE. As a result, for metal ions, *k*_i_ values between water and the plasticized PVC membrane should decrease compared to the corresponding values in the water–pure o-NPOE system, which was actually observed. For instance, the *k*_i_ value for cesium ions obtained in the present work is about one order of magnitude lower than values given in other works. In the case of anions, the situation is the opposite: the solvation energies of anions depend greatly upon the strength of hydrogen bonds formed. In PVC molecules, the hydrogen atoms of the ClC-H groups carry an excess positive charge, which makes this possible. Naturally, the energy of the hydrogen bonds increases with an increase in the electron density in anions. Therefore, the *k*_i_ values for hydrophilic anions between water and plasticized PVC membrane should be higher than those between water and pure o-NPOE, which agrees with the data in the present work and in [28]. Accordingly, the differences between the *k*_i_ values for TPB^−^ and more hydrophilic anions (from ClO_4_^−^ to Cl^−^) obtained by us are about an order of magnitude smaller than those presented in [28]. The above considerations confirm the reliability of the values of single-ion partition coefficients obtained by us.

Knowing the single-ion partition coefficients makes it possible to calculate the values of the coextraction constants of the corresponding pairs of ions. This parameter is a characteristic of the overall lipophilicity of the electrolyte and is of great practical importance, because it determines the value of the upper detection limit of neutral carrier-based ISEs [12,13,14,15,16], as well as the lifetime of indicator and reference electrodes [18,47], when their membranes contain moderately lipophilic ionic liquids or dyes that can gradually leach into the test solution. Table 3 shows the values of the logarithms of the coextraction constants calculated from the data given in Table 2, as well as those determined by the sandwich membrane method from potentiometric data [21].

It can be seen from the data presented that the calculated values of the coextraction constants correlate relatively well with each other (in the span of more than 25 decimal places, the maximum difference rarely exceeds 2 decimal places). This is partly due to the specifics of the voltammetric methods used, when the error in determining the zero potential has an opposite effect on the values of the calculated Gibbs energies of the cation and anion transfer.

It follows from Figure 2 that the values of the coextraction constants calculated from our data correlate almost perfectly with the values obtained from the data given in [28], although the values of single-ion partition coefficients in some cases differ significantly, especially for the most hydrophilic cations and anions. However, it should be noted that such a good correlation is, to some extent, a coincidence caused by the medium effect, upon changing pure o-NPOE for a plasticized PVC membrane, which is opposite for cations and anions. At the same time, it should be noted that, in some cases (for example, when justifying the choice of an ionic liquid that ensures the independence of the potential of the reference electrode from the composition of the solution under study, or for regulating the operating range of optodes), not only the co-extraction constants but also the single-ion partition coefficients are of decisive importance [17,18].

## 5. Conclusions

The integrated approach for experimental determination of single-ion partition coefficients, which combines the potentiometric method and partitioning technique to study the equilibrium state between water and PVC membrane plasticized with o-NPOE, was substantiated, and the corresponding values for nine cations and nine anions were determined. The constructed scale of single-ion partition coefficients can be easily extended, primarily with respect to singly charged anions. If higher quaternary ammonium salts with four long-chain substituents at the nitrogen atom are used as an ion exchanger, then the formation of ion associates can be neglected and the determination of single-ion partition coefficients is reduced to a routine procedure for determining unbiased selectivity coefficients. The problem becomes somewhat more complicated in the case of cations, because their association with tetraphenylborate derivatives, which are traditionally used as cation exchangers, cannot always be neglected. However, the values of the association constants can be easily estimated by the reference ion method. Obtaining such data is of great interest for improving the technique of optical (using ion-selective optodes) and potentiometric measurements. In particular, the possibility of tuning the analytical response of optodes by introducing an electrolyte with the required ratio of the cation and anion lipophilicities into the test solution has been demonstrated [18]. Of no less interest is the a priori rational choice of ionic liquids used to stabilize the boundary potential for arranging solution-free liquid junctions in miniature potentiometric cells [17,18,48,49]. It should be mentioned that the thermodynamic criteria for controlling the boundary potential at the membrane–solution interface, based on the lipophilicity characteristics of cations and anions of some auxiliary electrolyte capable of limited distribution between the solution and the membrane, have been previously developed. However, lack of reliable experimental data may be a constraining factor for the implementation of these ideas. We believe that the feasible and direct approach developed in the present article will be fruitful and will facilitate progress in this area.

## Figures and Tables

**Figure 1 membranes-12-01019-f001:**
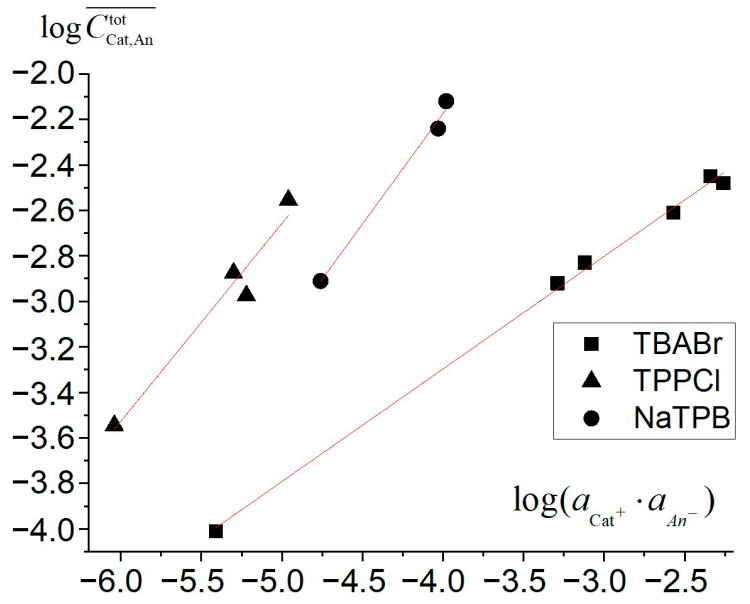
The bi-logarithmic dependences of total salt concentrations in the membrane phase upon electrolyte activity in the solution. The fitted lines are as follows: y = 0.495x − 1.273 for TBABr; y = 0.87x + 1.74 for TPPCl; y = 0.97x + 1.75 for NaTPB.

**Figure 2 membranes-12-01019-f002:**
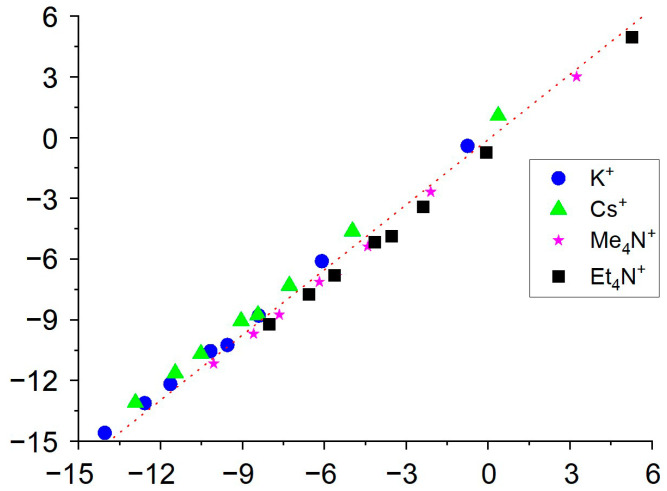
Correlation between coextraction constants calculated from [28] (Y-axis) and obtained in this paper (X-axis). Anions from left to right are as follows: Cl^−^, Br^−^, NO_3_^−^, I^−^, SCN^−^, ClO_4_^−^, Pic^−^, TPB^−^. The equation of the correlation line is y = 1.07x − 0.10.

**Table 1 membranes-12-01019-t001:** Unbiased potentiometric selectivity coefficients for cation- and anion-exchange membranes and corresponding ion-exchange constants.

Cation	$\log K_{{\mathrm{Cat}}^{+}{,\mathrm{TPP}}^{+}}^{\mathrm{Pot}}$	$\log K_{{\mathrm{Cat}}^{+}{,\mathrm{TPP}}^{+}}^{\mathrm{Exch}}$ ***	Anion	$\log K_{{\mathrm{An}}^{-}{,\mathrm{TPB}}^{-}}^{\mathrm{Pot}}$ , $\log K_{{\mathrm{An}}^{-}{,\mathrm{TPB}}^{-}}^{\mathrm{Exch}}$ **
Li^+^	13.44 ± 0.05	13.99 *	Cl^−^	13.28 ± 0.05
Na^+^	12.64 ± 0.05	13.19	Br^−^	11.83 ± 0.07
K^+^	10.92 ± 0.05	11.49	C_6_H_5_SO_3_^−^	10.90 ± 0.06
Rb^+^	10.35 ± 0.06	10.94	NO_3_^−^	10.89 ± 0.08
Cs^+^	9.74 ± 0.05	10.37	I^−^	9.41 ± 0.04
TMA^+^	7.62 ± 0.05	7.50	SCN^−^	8.80 ± 0.05
TEA^+^	5.69 ± 0.05	5.47	ClO_4_^−^	7.64 ± 0.07
TBA^+^	1.68 ± 0.08	1.46	Pic^−^	5.34 ± 0.06
TPP^+^	≡0	≡0	TPB^−^	≡0

(*) Estimated value for which the equality of ion association constants of TClPB^−^ with sodium and lithium cations was assumed. (**) The confidence interval represents the reproducibility of the potentiometric measurements and does not account for the electrode’s response distortion caused by the diffusional potential across the membrane. (***) Taking into account the uncertainty in association constant values, the uncertainty in ion exchange constants does not exceed 0.2. TMA^+^ is tetramethylammonium cation; TEA^+^ is tetraethylammonium cation; TBA^+^ is tetrabutylammonium cation; TPP^+^ is tetraphenylphosphonium cation; Pic^−^ is picrate anion.

**Table 2 membranes-12-01019-t002:** The single-ion partition coefficients from water to o-NPOE-PVC membrane (this paper, in bold) and to o-NPOE (literature data).

Ion	$logk_{i}$, Water—o−NPOE *					
	This Paper	[25] ^a^	[25] ^b^	[29]	[32]	[28]
Li^+^	**−8.62**					
Na^+^	**−7.82**				−6.36	
K^+^	**−6.12**					−5.81
Rb^+^	**−5.57**					
Cs^+^	**−5.00**	−4.03	−3.68			−4.32
TMA^+^	**−2.13**		−1.88	3.29	−1.89	−2.39
TEA^+^	**−0.10**	−0.46	−0.46	−0.46	−0.82	−0.44
TPA^+^			1.52		1.37	1.58
TBA^+^	**3.91**	4.08			3.15	
TPP^+^	**5.37**				4.35	
TPAs^+^		5.31				5.40
TPB^−^	**5.37**	5.31	5.52		4.35	5.40
Pic^−^	**0.03**		0.65	−0.66		−0.30
ClO_4_^−^	**−2.28**	−2.70	− 2.33	−2.65	−1.86	−2.99
SCN^−^	**−3.43**				−3.36	−4.44
I^−^	**−4.05**				−3.70	−4.74
NO_3_^−^	**−5.52**				−3.82	−6.36
C_6_H_5_SO_3_^−^	**−5.53**					−6.16
Br^−^	**−6.46**				−5.92	−7.31
Cl^−^	**−7.92**				−7.55	−8.78

* If the original source operated in terms of Δ*G*^0^, the *k*_i_ values were calculated by the formulae: $\log k_{i}=\frac{-\Delta G{}^{\circ}}{2.303RT}$
, where *T* was taken from the experimental part of the corresponding paper; ^a^—from solubility data; ^b^—from voltammetry. TPA^+^ is tetrapropylammonium cation; TPAs^+^ is tetraphenylarsonium cation; TPB^−^ is tetraphenylborate anion.

**Table 3 membranes-12-01019-t003:** Calculated coextraction constants for various electrolytes. Numbers in bold represent the results of the present article.

	Li^+^	Na^+^	K^+^	Cs^+^	TMA^+^	TEA^+^	TBA^+^	TPP^+^
TPB^−^	**−3.25**	**−2.45** −2.01 [32]	**−0.75** −0.41 [28]	**0.37** 1.28 [25] ^a^ 1.84 [25] ^b^ 1.08 [28]	**3.24** 3.64 [25] ^b^ 2.46 [32] 3.01 [28]	**5.27** 4.85 [25] ^a^ 5.06 [25] ^b^ 3.53 [32] 4.96 [28]	**9.28** 9.39 [25] ^a^ 7.50 [32]	**10.74** 8.70 [32]
Pic^−^	**−8.59**	**−7.79**	**−6.09** −6.11 [28]	**−4.97** −3.03 [25] ^b^ −4.62 [28]	**−2.10** −1.23 [25] ^b^ −2.63 [29] −2.69 [28]	**−0.07** 0.19 [25] ^b^ −1.12 [29] −0.74 [28]	**3.94**	**5.4**
ClO_4_^−^	**−10.9**	**−10.10** −8.22 [32] −8.4 [21]	**−8.40** −8.80 [28] −7.2 [21]	**−7.28** −6.73 [25] ^a^ −6.01 [25] ^b^ −7.31 [28]	**−4.41** −4.21 [25] ^b^ −0.64 [29] −3.75 [32] −5.38 [28]	**−2.38** −3.16 [25] ^a^ −2.79 [25] ^b^ −3.11 [29] −2.68 [32] −3.43 [28]	**1.63** 1.38 [25] ^a^ 1.29 [32]	**3.09** 2.49 [32]
SCN^−^	**−12.05**	**−11.25** −9.72 [32] −9.8 [21]	**−9.55** −10.25 [28] −8.6 [21]	**−8.43** −8.76 [28]	**−5.56** −5.25 [32] −6.83 [28]	**−3.53** −4.18 [32] −4.88 [28]	**0.48** −0.21 [32]	**1.94** 0.99 [32]
I^−^	**−12.67**	**−11.87** −10.2 [21]	**−10.17** −10.55 [28] −9.0 [21]	**−9.05** −9.06 [28]	**−6.18** −5.59 [32] −7.13 [28]	**−4.15** −4.52 [32] −5.18 [28]	**−0.14** −0.55 [32]	**1.32** 0.65 [32]
NO_3_^−^	**−14.14**	**−13.34** −10.18 [32] −11.5 [21]	**−11.64** −12.17 [28] −10.3 [21]	**−10.52** −10.68 [28]	**−7.65** −5.71 [32] −8.75 [28]	**−5.62** −4.64 [32] −6.80 [28]	**−1.61** −0.67 [32]	**−0.15** 0.53 [32]
Br^−^	**−15.08**	**−14.28** −12.28 [32] −12.4 [21]	**−12.58** −13.12 [28] −11.2 [21]	**−11.46** −11.63 [28]	**−8.59** −7.81 [32] −9.70 [28]	**−6.56** −6.74 [32] −7.75 [28]	**−2.55** −2.77 [32]	**−1.09** −1.57 [32]
Cl^−^	**−16.54**	**−15.74** −13.91 [32] −13.6 [21]	**−14.04** −14.59 [28] −12.4 [21]	**−12.92** −13.10 [28]	**−10.05** −9.44 [32] −11.17 [28]	**−8.02** −8.37 [32] −9.22 [28]	**−4.01** −4.40 [32]	**−2.55** −3.20 [32]

^a^—from solubility data; ^b^—from voltammetry.

## Data Availability

Not applicable.

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
