# Peer review of "Determination of Single-Ion Partition Coefficients between Water and Plasticized PVC Membrane Using Equilibrium-Based Techniques"

_membranes, 2022, doi:10.3390/membranes12101019_

Round 1

Reviewer 1 Report

This manuscript reports the determination of single-ion partition coefficient directly using the equilibrium-based method. This is a very fundamental research, which impressed me a lot. Most early studies have used the voltammetric technique to determine the partition coefficients (biased coefficient). This work offered unbiased results based on primal distribution method. From Figure 2, it can be concluded the measures results are correlated with previous method. I have no more comments for this work, just a suggestion that is double check the format of the equations described in the text. These equations should be presented in the journal required format, for example, the font, the subscript and superscript. Additionally, the format for the numbers, ions in the text should be unified according to the journal requirements. The TOC is better using a higher resolution picture. The symbol for ions in Table D2, Table E1 and Table E2 should be given (cations (+) or anions (–). I recommend publication of this work.

Author Response

The authors express their deep gratitude to the Reviewer for the thorough consideration of the manuscript and valuable suggestions.

Responses to the Reviewer's notes:

  • These equations should be presented in the journal required format, for example, the font, the subscript and superscript.

Answer to the note:

Thank you for the note. All the formulas and symbol definitions were corrected with symbols in italics and words / numbers in non-italic font using MathType add-on as recommended by the Journal Guideline.

  • Additionally, the format for the numbers, ions in the text should be unified according to the journal requirements.

Answer to the note:

Thank you for the note. All the numbers and ion names were unified in the text. See 1) Abstract, line 4 2) P.2 lines 8 and 10; 3) P.2 Par. 2 line 7; 4) P.2 Brief theoretical background, lines 2 and 3 5) P.3 lines 4 and 9 6) P.3. Par.3 line 1; 7) P.3, last Par lines 2 and 3; 8) P.4.line 5; 9) P.6 Results and Discussion, lines 5 and 9; 10) Fig.1;11) P.8 lines 5, 7 ,9 and 11 12) Table 1 13) P.11 line 2; 14) P.11, Par.2, lines 4 from the end of the Par. 15) Table.3 16) Table A.1 and 1st line after 17) line 5 after eq A.1; 16) lines 1,2 and 5 after Table A2;17) line 1 after Table A3.

  • The TOC is better using a higher resolution picture.

Answer to the note:

Thank you for the note. The picture is updated using a higher resolution.

  • The symbol for ions in Table D2, Table E1 and Table E2 should be given (cations (+) or anions (–)).

Answer to the note:

Thank you for the note. The required symbols are added in Table D2, Table E1 and Table E2.

Reviewer 2 Report

The manuscript aims at determining single-ion partition coefficients. Although it is well written, it needs some minor corrections.

The English should be polished in the entire text as at some places it could be improved typically “in the Figure” and “in the Table” which would be correct without “the”. In page 10, line 331, I recommend to write instead “data for cations more hydrophilic than cesium and anions more hydrophilic than perchlorate” the following: “data for cations which are more hydrophilic than cesium and for anions which are more hydrophilic than perchlorate”. Furthermore, instead of “ion activities product” write “product of ion activities”. There are word duplications at several places.

Empty rows should be inserted between the captions of figures and tables to separate them from the main text as well as after figures and tables.

The whole Table 2 should be on one page.

I advise the completing of captions of Table 1 and 2 with the list of abbreviations of untrivial ion names.

Why did the authors use potentiometric titration for concentrations above 5∙10-3 M and direct potentiometry for concentrations below it? Was it due to dynamic range of TBA+-selective electrode (upper detection limit)?

In an equation (page 3, line 91), the second bracket ] is at wrong place, it should be after RT as the whole expression is in the exponent.

The captions of Table E1 and E2 are not clear because of “for cations to lithium cation” and “for anions to TPB-anion”.

Are the measured and calculated results appropriate for establishing upper detection limits for membranes with different ionic pairs?

After these corrections I recommend the work for publication.

Author Response

The authors express their deep gratitude to the Reviewer for the thorough consideration of the manuscript and valuable suggestions.

  • The English should be polished in the entire text as at some places it could be improved typically “in the Figure” and “in the Table” which would be correct without “the”.

Answer to the note:

Thank you for the note. The changes are performed at 1) P.1, Abstract, line 13 2) P.1 Introduction, line 4 3) P.2 Par.3, last sentence 4) P.2 Par.4 line.7 5) P.3 lines 9 and 12 6) P.4 line 10 7) P.6 Results and Discussion line 6 8) Page 7 Par.1 9) P.8 line 6 10) P. 9, before Table 2 11) P. 13 and 14 Conclusion lines 19-23 12) P.19 Appendix E line 10.

  • In page 10, line 331, I recommend to write instead “data for cations more hydrophilic than cesium and anions more hydrophilic than perchlorate” the following: “data for cations which are more hydrophilic than cesium and for anions which are more hydrophilic than perchlorate”. Furthermore, instead of “ion activities product” write “product of ion activities”.

Answer to the note:

Thank you for the note. The changes are performed at 1) Page 7 lines 4,8,25 2) Page 10 after Table 2 lines 3 and 4.

  • There are word duplications at several places.

Answer to the note:

Thank you for the note. The duplications were corrected at 1) P.1 Introduction, line 4 2) P.2 Par.4 Line 7 3) P.4. Par.3 line 4; 4) P.4 Materials and Methods line 14 6) P.5 Par.5 lines 7 and 9.

  • Empty rows should be inserted between the captions of figures and tables to separate them from the main text as well as after figures and tables.

Answer to the note:

Thank you for the note. The changes are performed for all the figures and tables.

  • The whole Table 2 should be on one page.

Answer to the note:

Thank you for the note. Table 2 is corrected.

  • I advise the completing of captions of Table 1 and 2 with the list of abbreviations of untrivial ion names.

Answer to the note:

Thank you for the note. Captions of Table 1 and 2 are completed with abbreviations.

  • Why did the authors use potentiometric titration for concentrations above 5∙10-3M and direct potentiometry for concentrations below it? Was it due to dynamic range of TBA+-selective electrode (upper detection limit)?

Answer to the note:

The potentiometric titration is more accurate method for high concentrations, however it does not allow to accurately measure low concentrations. Direct potentiometry is less accurate, but it is possible to measure low concentrations (≈10-5 Ðœ) with a TBA+-selective electrode.

  • In an equation (page 3, line 91), the second bracket ] is at wrong place, it should be after RT as the whole expression is in the exponent.

Answer to the note:

Thank you for the note. The equation is corrected (P. 3, line 4).

  • The captions of Table E1 and E2 are not clear because of “for cations to lithium cation” and “for anions to TPB-anion”.

Answer to the note:

Thank you for the note. The captions of Table E1 and E2 are corrected “for cations (primary ion is sodium cation)” and “for anions (primary ion is chloride-anion)”

  • Are the measured and calculated results appropriate for establishing upper detection limits for membranes with different ionic pairs?

Answer to the note:

Yes, this point is explained at Page 11” Knowing the single-ion partition coefficients makes it possible to calculate the values of the coextraction constants of the corresponding pairs of ions. This parameter is a characteristic of the overall lipophilicity of the electrolyte and is of great practical importance, because it determines the value of the upper detection limit of neutral carriers-based ISEs [12-16]…”